# Distributed Optimization of Multi-Microgrid Integrated Energy System with Coordinated Control of Energy Storage and Carbon Emissions

**Linjun Shi** [1], **Zimeng Cen** [1,*], **Yang Li** [1], **Feng Wu** [1], **Keman Lin** [1] and **Dongmei Yang** [2]

[1] School of Electrical and Power Engineering, Hohai University, Nanjing 211100, China; eec@hhu.edu.cn (L.S.); eeliyang@hhu.edu.cn (Y.L.); wufeng@hhu.edu.cn (F.W.); linkeman@hhu.edu.cn (K.L.)
[2] State Key Laboratory of Smart Grid Protection and Control, Nari Group Corporation, Nanjing 211106, China; yangdongmei@sgepri.sgcc.com.cn
* Correspondence: 221606030081@hhu.edu.cn

**Abstract:** The mutual optimization of a multi-microgrid integrated energy system (MMIES) can effectively improve the overall economic and environmental benefits, contributing to sustainability. Targeting a scenario in which an MMIES is connected to the same node, an energy storage coordination control strategy and carbon emissions management strategy are proposed, and an adaptive step-size method is applied to improve the distributed optimization of MMIESs based on the alternating direction multiplier method (ADMM). Firstly, the basic framework of MMIESs is established, and a coordinated control strategy limiting the time of charge and the discharge of the battery storage system (BSS) is proposed. Then a multi-objective optimization model based on operating and environmental cost is formulated. Considering that different microgrids may be managed by different operators and a different convergence speed of multi-objective optimization iteration, an adaptive step-size distributed iterative optimization method based on ADMM is used, which can effectively reduce the cost and protect the privacy of each microgrid. Finally, a system composed of three microgrids is taken as an example for simulation analysis. The results of distributed optimization are accurate, and the proposed coordinated control strategy can effectively enhance the revenue of ESS, which verifies the effectiveness of the proposed method.

**Keywords:** multi-microgrid integrated energy system (MMIES); energy storage coordination control strategy; alternating direction multiplier method (ADMM); carbon emissions

## 1. Introduction

With the intensification of energy shortages and environmental degradation, there is an urgent need for reasonable means to improve the utilization efficiency of various types of energy and reduce carbon emissions. The issue of sustainable development in society has become very important. The mutual coupling of cold, heat, electricity, and gas energy in the integrated energy system (IES), is widely used in the electric power industry and has become one of the most important ways to solve these problems [1].

The IES contains distributed wind power, photovoltaic and other renewable energy power generation units, which have a certain randomness and uncontrollability. It may be that some renewable energy generation of IES is more than the load demand, and there might be a phenomenon of curtailing wind and solar power, resulting in a waste of resources. Sometimes the IES is in short supply of energy and needs to purchase power from the power grids. If a reasonable connection between two IESs is established, forming a multi-microgrid integrated energy system (MMIES), the power exchange between each of the microgrids can effectively solve the problem of balancing supply and demand, and reducing the cost. Thus, how to optimize the scheduling of an MMIES has become an important issue at present [2–4].

The optimized MMIES scheduling models can be divided into two categories: centralized and distributed models [3]. Although the centralized optimization algorithm is simple and fast, it needs to transmit a lot of data from all of the microgrids in the system to the control center for unified scheduling, resulting in a huge amount of traffic, and weak flexibility and stability [4]. As a result, distributed algorithms are used to ensure the density and security of each microgrid. The mainstream distributed algorithms used are based on the Lagrangian relaxation (LR) method, including optimal condition decomposition (OCD) [5–7] and generalized benders decomposition (GBD) [8,9], which can realize a completely distributed solution for multi-region scheduling problems. Other distributed algorithms in use are based on the augmented relaxation description (ALD) method, including the auxiliary problem principle (APP) [10–12] and analytical target cascading (ATC) [13–15], which can improve the convergence performance of the high standard LR method [16].

The alternating direction multiplier method (ADMM) is also a kind of ALD method. Compared with other methods, ADMM can deal with complex models effectively, and its convergence and convergence speed have been proven, providing the basic framework for MMIESs. In the basic ADMM scheduling framework, Mohiti M et al. [17] took economic cost as the scheduling objective function, Chen L et al. [18] established an optimal two-layer multi-timescale scheduling model based on short-term forecast data, and Cheng S et al. [19] put more emphasis on price policy, establishing a Nash bargaining cooperative game model to minimize the operating cost of each microgrid.

The above studies considered the privacy and security of a single microgrid and adopted a distributed algorithm for optimization, where most of them had economy as the optimization target. In addition to the minimum operating cost, exergy optimization takes the minimum amount of wind and light used [20], the minimum annual operating cost of a shared battery storage system (BSS) [21], and the network losses and voltage quality as the objective functions, while environmental considerations are rarely taken into account. In addition, some scholars have studied BSS in MMIESs. Safari et al. [22] studied the impact of the clean output of hydrogen storage systems and fuel cells on system uncertainty. Chuanshen et al. [23] utilized the mobile energy storage characteristics of electric vehicles to improve the performance of MMG systems. However, there are few studies on the control of the charging and discharging frequency of energy storage devices in MMIESs. If the optimization is directly based on the existing control strategy, there may be frequent charging and discharging, which will adversely affect the service life of the storage devices.

Therefore, on the basis of establishing the basic framework of MMIESs containing cold, heat, electricity, and gas, this paper proposes an energy storage coordination control strategy that limits the number of charging and discharging times and takes the operating cost and the environmental cost based on carbon dioxide emissions as the optimization targets. Considering that different microgrids may be managed by different operators and to enhance the convergence speed of multi-objective optimization iteration, an adaptive step-size solution method is proposed to improve ADMM distributed iterative optimization, which can effectively reduce the cost and protect the privacy of each microgrid. Finally, an MMIES composed of three microgrids is taken as an example for simulation analysis. The simulation results show that the distributed optimization results are consistent with the centralized optimization results, and the proposed coordinated control strategy can effectively enhance the revenue of BSS, verifying the effectiveness of the proposed method.

## 2. Multi-Microgrid Integrated Energy Systems (MMIESs)

### 2.1. Architecture of MMIES Considering Carbon Emissions

The MMIES considered in this paper can be divided into two layers. The upper management system contains distribution points of electricity and natural gas and the carbon emissions management system that sets the carbon emissions strategy. The lower system

contains regional microgrids, the control system and the load under the management of the regional operator.

The system consists of three microgrids connected at the same distribution point of electricity and natural gas, but different microgrids are managed and controlled by different regional operators to provide different regional users. Residential, commercial, industrial, and other regional users have different load characteristics. When several microgrids are connected, there are mutual benefits between them. They can make the surplus of renewable energy power generation in one microgrid supply the power to other microgrids, reducing the energy purchase from the power grid, thereby reducing the overall operating cost and avoiding energy waste. Each microgrid is connected in pairs through power channels. Considering that the distance between microgrids is generally several kilometers, the heat pipeline will cause great heat loss when transmitting at this distance. Therefore, the thermal energy interaction between microgrids is not considered. Each microgrid can exchange power with an external distribution network. The natural gas required for each microgrid is purchased from the external natural gas network, and the sale of natural gas to the natural gas network is not considered. The power flow interaction of the system established in this paper is shown in Figure 1.

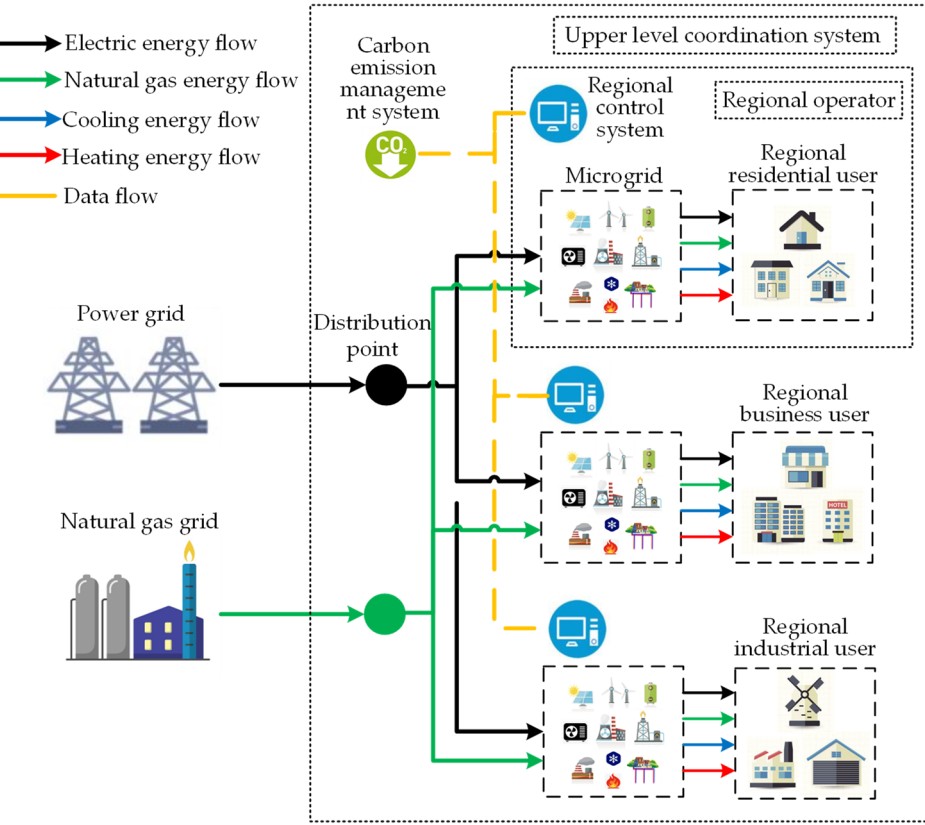

**Figure 1.** Basic architecture of MMIES.

## *2.2. Models for Internal Structure of Single Microgrid Integrated Energy System (MIES)*

The MIES is mainly composed of energy network, energy conversion link, energy storage link, and end users. In the comprehensive model established in this paper, the energy supply network includes electricity, gas, cooling, and heating networks. Through the reasonable output of each piece of equipment in the system, the supply and demand balance of electricity, gas, cooling, and heating is reached. The equipment for energy conversion and storage and the energy-flow relationships are shown in Figure 2. The specific device model formulas are as follows, sourced from Jianfeng et al. [24].

1.  Gas Turbine (*GT)*

$$\begin{cases} P_{GT,t} = \eta_{GT} \times G_{GT,t} \times q_{\lg} \\ 0 \leq P_{GT,t} \leq P_{GT,t}^{\max} \end{cases} \tag{1}$$

where $P_{GT,t}$ is the power generation of *GT*; $\eta_{GT}$ is the power generation efficiency of *GT*; $G_{GT,t}$ is the consumption of natural gas; and $q_{\lg}$ is the low calorific value of natural gas.

2.  Gas Boiler (*GB*)

$$\begin{cases} H_{GB,t} = \eta_{GB} \times G_{GB,t} \times q_{lq} \\ 0 \leq H_{GB,t} \leq H_{GB,t}^{\max} \end{cases} \tag{2}$$

where $H_{GB,t}$ is the heat produced by *GB*; $\eta_{GB}$ is operating efficiency; and $G_{GB,t}$ is the natural gas consumption.

3.  Absorption Chiller (*AC*)

$$\begin{cases} C_{AC,t} = CO_{AC} \times H_{AC,t} \\ H_{AC,t} = (1 - \eta_{GT}) \times G_{GT} \times q_{\lg} \times \alpha_{AC} \\ 0 \leq C_{AC,t} \leq C_{AC}^{\max} \end{cases} \tag{3}$$

where $C_{AC,t}$ is the cooling capacity of *AC*; $CO_{AC}$ is the refrigeration coefficient of *AC*; $H_{AC,t}$ is the input heat power of *AC*; and $\alpha_{AC}$ is the residual heat distribution coefficient.

4.  Heat-recovery Boiler (*HB*) The *HB* recovers a portion of the heat from the *GT* to meet the heating demand of users.

$$\begin{cases} H_{HB,t} = \eta_{HB} \times H_{HB\_in,t} \\ H_{HB\_in,t} = (1 - \eta_{GT}) \times G_{GT,t} \times q_{\lg} \times (1 - \alpha_{AC}) \\ 0 \leq H_{HB,t} \leq H_{HB}^{\max} \end{cases} \tag{4}$$

where $H_{HB,t}$ is the heat produced by *HB* and $H_{HB\_in,t}$ is the heat energy.

5.  Electric Chiller (*EC*)

$$\begin{cases} C_{EC,t} = CO_{EC} \times P_{EC,t} \\ 0 \leq C_{EC,t} \leq C_{EC}^{\max} \end{cases} \tag{5}$$

where $P_{EC,t}$ is the energy consumption of *EC*.

6.  Ground Source Heat Pump (*GSHP*)

$$\begin{cases} H_{GSHP,t} = CO_{GSHP\_heat} \times P_{GSHP\_heat,t} \\ C_{GSHP,t} = CO_{GSHP\_cool} \times P_{GSHP\_cool,t} \\ 0 \leq P_{GSHP\_cool,t} \leq u_{GSHP} \times P_{GSHP\_cool}^{\max} \\ 0 \leq P_{GSHP\_heat,t} \leq (1 - u_{GSHP}) \times P_{GSHP\_heat}^{\max} \\ (1 - u_{GSHP})u_{GSHP} = 0 \end{cases} \tag{6}$$

where $H_{GSHP,t}$ is the heat energy added by *GSHP*; $C_{GSHP,t}$ is the cooling energy added by *GSHP*; $P_{GSHP\_heat,t}$ is the electricity consumption for heating; and $P_{GSHP\_cool,t}$ is the electricity consumption for cooling.

7.  Power to Gas (*PtG*) Equipment

$$\begin{cases} G_{PtG,t} = \eta_{PtG} \times P_{PtG,t} / q_{hg} \\ 0 \leq P_{GT,t} \leq P_{GT,t}^{\max} \end{cases} \tag{7}$$

where $q_{hg}$ is the high calorific value of natural gas.

8.  Internal Power Balance of Microgrid

The MIES needs to meet the constraints of the energy balance of the four subsystems of electrical, gas, cooling, and heating during operation, which means that the energy supply of the subsystems reaches a real-time balance with the needs of users.

$$P_{PV,t} + P_{WT,t} + P_{SB\_out,t} + P_{GT,t} + P_{grid,t} - P_{SB\_in,t} - P_{EC,t} - P_{GSHP\_heat,t} - P_{GSHP\_cool,t} - P_{PtG,t} - P_{sold,t} = P_{load,t} \tag{8}$$

$$G_{NG,t} - G_{GT,t} - G_{GB,t} + G_{PtG,t} = G_{load,t} \tag{9}$$

$$C_{AC,t} + C_{EC,t} + C_{GSHP,t} = C_{load,t} \tag{10}$$

$$H_{GB,t} + H_{HB,t} + H_{GSHP,t} = H_{load,t} \tag{11}$$

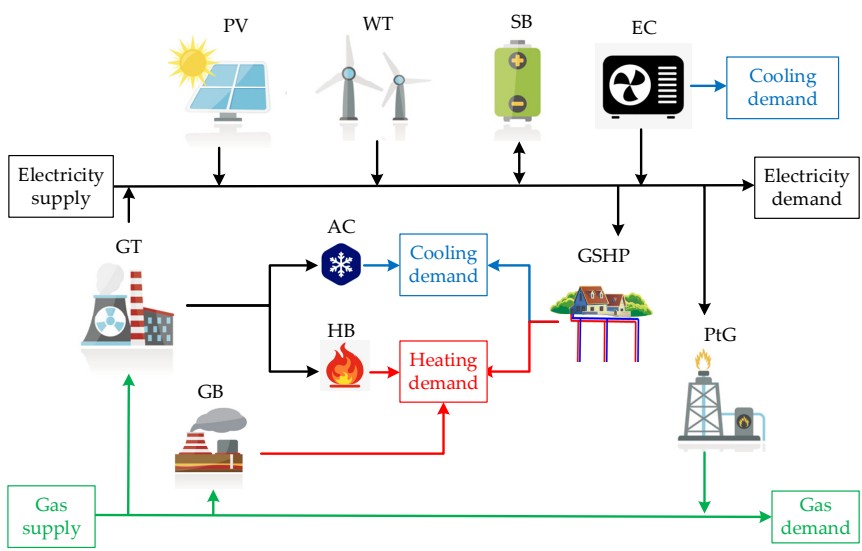

**Figure 2.** Basic architecture of MMIES.

9. Power Interaction between Microgrid and Distribution Network

In order to ensure adequate safety and reliability, reasonable constraints should be added to the process of purchasing electricity from the grid or selling electricity to the grid:

$$\begin{cases} 0 \leq P_{grid,t} \leq u_g \times P_{grid}^{\max} \\ 0 \leq P_{sold,t} \leq (1 - u_g) \times P_{grid}^{\max} \\ (1 - u_g)u_g = 0 \end{cases} \tag{12}$$

## 3. Energy Storage Coordination Control Strategy

### 3.1. Energy Storage Device Model

The energy stored in the energy storage link can be electricity, natural gas, cold energy, and heat energy. In the model of this paper, the devices that store electricity are considered. The technology of the battery storage system (BSS) is mature and widely used, so the use of BSS as an energy storage device can play the role of peak cutting and valley filling, providing an economic solution as well as environmental protection.

The state of the energy storage device is the state of the available energy in BSS, which is described by the state of charge (SOC). The SOC can accurately display the remaining capacity of BSS, and its mathematical model is:

$$SOC = E_{BSS}/E_{BSS\_r} \tag{13}$$

Charge:

$$SOC_{BSS,t} = (1 - \sigma_{BSS}) \times SOC_{BSS,t-1} + \eta_{BSS\_in} \times P_{BSS\_in,t} \times \Delta t/E_{BSS\_r} \tag{14}$$

Discharge:

$$SOC_{BSS,t} = (1 - \sigma_{BSS}) \times SOC_{BSS,t-1} - \eta_{BSS\_out} \times P_{BSS\_out,t} \times \Delta t/E_{BSS\_r} \tag{15}$$

which simultaneously needs to satisfy the following:

$$\begin{cases} 0 \leq P_{BSS\_in,t} \leq u_{BSS,in,t} \times P_{BSS}^{\max} \\ 0 \leq P_{BSS\_out,t} \leq u_{BSS,out,t} \times P_{BSS}^{\max} \\ SOC_{BSS}^{\min} \leq SOC_{BSS,t} \leq SOC_{BSS}^{\max} \\ u_{BSS,in,t} + u_{BSS,out,t} \leq 1 \end{cases} \tag{16}$$

where $E_{BSS}$ is the current energy storage capacity of *BSS*; $E_{BSS\_r}$ is the maximum energy storage capacity; $SOC_{BSS,t}$ and $SOC_{BSS,t-1}$ are *SOC* at time t and time t−1; $\sigma_{BSS}$ is the power consumption rate of *BSS*; $\eta_{BSS\_in}$ is the conversion efficiency of the energy absorbed by *BSS*; $\eta_{BSS\_out}$ is the conversion efficiency of the energy supplied by *BSS*; $P_{BSS\_in,t}$ is the power absorbed by *BSS*; and $P_{BSS\_out,t}$ is the power supplied by *BSS*.

### 3.2. Cost Indicators for BSS Construction and Operation

In order to clearly determine the benefits of the coordinated control strategy for *BSS* grouping on system operation, this paper proposes a cost indicator function that considers the construction cost and the operation and maintenance cost of *BSS*.

$$f_{BSS} = f_{BSS\_INV} + f_{BSS\_OM} \tag{17}$$

$$f_{BSS\_INV} = \frac{r(1+r)^y}{365[(1+r)^y - 1]}(\alpha E_{BSS}^{Cap} + \beta P_{BSS}^{Cap}) \tag{18}$$

$$f_{BSS\_OM} = \gamma \sum_{t=1}^{T} (P_{BSS,in,n,t} + P_{BSS,out,n,t}) \tag{19}$$

where $f_{BSS}$ is the *BSS* cost indicator function; $f_{BSS\_INV}$ and $f_{BSS\_OM}$, respectively, represent the construction cost and the operation and maintenance cost of *BSS*; $r$ is the discount rate; $y$ is the service life of *BSS*; $E_{BSS}^{Cap}$ and $P_{BSS}^{Cap}$ are the energy capacity and power capacity of BSS; $\alpha$ and $\beta$ are the unit energy capacity cost and the unit power capacity cost; and $\gamma$ is the unit cost for energy storage charging and discharging.

### 3.3. Energy Storage Coordination Control Strategy Modle

Energy storage devices can absorb renewable energy generation and play the role of peak cutting and valley filling. Existing literature on BSS in MMIESs mainly focuses on the impact of BSS on system uncertainty [22], or the improvement of system performance by similar energy storage devices [23]. It is overlooked that in an MMIES, multiple BSSs working together can bring benefits to both the multi-microgrid system and the BSSs. In a single MIES, in order to minimize economic costs, energy storage is usually in a state of repeated charge and discharge, which will adversely affect the capacity and service life of energy storage devices. The price of energy storage equipment is high, and if it is not fully utilized, it will further harm the economy and the environmental protection of the entire system.

However, in MMIESs there can be multiple energy storage devices, and formulating a reasonable energy storage coordination control strategy can effectively solve the above problems. When multiple energy storage devices of the same type are used in parallel in a system, they can be selectively allowed to participate in power and energy distribution according to certain rules and the system's forecast or real-time energy storage demand, so as to improve the efficiency of the ESS. The life of a battery is related to the number of charge and discharge cycles, peak current, and temperature.

In this paper, the MMIES contains multiple energy storage batteries. To prolong the life of the batteries as much as possible, a control strategy for the charge and discharge of energy storage equipment is developed for the system. Compared to a single microgrid system, there is mutual power between microgrids in MMIESs. When a single microgrid system lacks power, in addition to purchasing power from the grid, it can only be adjusted through BSS charging and discharging. In MMIESs, when a certain microgrid lacks power, it can also be adjusted through the mutual power provided by other microgrids. By combining

with a charging and discharging strategy, the burden of BSS in MMIESs can be effectively reduced. This strategy can control the number of single BSS charges and discharges per day within a certain range, without repeated charging and discharging. In addition, the state of charge at the beginning and end remains unified, so that the battery can be charged and discharged according to the cycle. The specific implementation is as follows:

$$u_{BSS,in,n,t} - u_{BSS,in,n,t-1} \leq z_{BSS,in,n,t} \tag{20}$$

$$u_{BSS,out,n,t} - u_{BSS,out,n,t-1} \leq z_{BSS,out,n,t} \tag{21}$$

$$\sum_{t=1}^{T} z_{BSS,in,n,t} \leq N \tag{22}$$

$$\sum_{t=1}^{T} z_{BSS,out,n,t} \leq N \tag{23}$$

These simultaneously need to satisfy the following:

$$u_{BSS,in,n,t} + u_{BSS,out,n,t} <= 1 \tag{24}$$

$$\sum_{t=1}^{T} P_{BSS,in,n,t} - \sum_{t=1}^{T} P_{BSS,out,n,t} = 0 \tag{25}$$

where $u_{BSS,in,n,t}$ represents the charging state of *BSS* in microgrid *n* and 1 represents charging; $u_{BSS,out,n,t}$ represents the discharging state of *BSS* in microgrid n, 1 represents discharging. $z_{BSS,in,n,t}$ and $z_{BSS,out,n,t}$ are 0–1 variables. *N* represents the maximum daily charging and discharging conversion times.

## 4. Optimization Scheduling of MMIES

### 4.1. Objective Function of Optimization Scheduling

1.  Economic Cost Objective Function

The optimal scheduling of an MMIES minimizes the sum of the operating costs of all MIESs in the system through energy management, so the objective function is expressed as follows:

$$\min \sum_{i=1}^{N} C_i^{MIES} \tag{26}$$

where *N* is the MIES assembly and $C_i^{MIES}$ is the operation cost of microgrid *i*, including the cost of electric energy use $C_{i,grid}$ and the cost of natural gas use $C_{i,gas}$. Income includes the income of the MIES selling electricity to the power grid $C_{i,load}$:

$$C_i^{MIES} = C_{i,grid} + C_{i,gas} - C_{i,load} \tag{27}$$

$$\begin{cases} C_{i,grid} = \sum\limits_{t=1}^{T} p_{grid,t} P_{i,grid,t} \Delta t \\ C_{i,gas} = \sum\limits_{t=1}^{T} p_{gas,t} G_{i,NG,t} \Delta t \\ C_{i,sold} = \sum\limits_{t=1}^{T} p_{sold,t} P_{i,sold,t} \Delta t \end{cases} \tag{28}$$

2.  Environmental Cost Objective Function

In response to the strategic goal of carbon peaking and carbon neutrality, environmental factors should also be considered in the operation optimization goal of MIES. The overall environmental and economic penalty objective function of MMIES is expressed as follows:

$$\min D = \lambda_D \left( D_{grid} + D_{GT} + D_{GB} \right) \tag{29}$$

$$\begin{cases} D_{grid} = \sum_{t=1}^{T} \varepsilon_{grid} P_{grid,t} \Delta t \\ D_{GT} = \sum_{t=1}^{T} \varepsilon_{GT} (P_{GT,t} + H_{AC,t} + H_{HA\_in,t}) \Delta t \\ D_{GB} = \sum_{t=1}^{T} \varepsilon_{GB} H_{GB,t} \Delta t \end{cases} \tag{30}$$

where $\lambda_D$ is the penalty factor of $CO_2$ emission; $D_{grid}$ is the amount of $CO_2$ produced by power grid; $D_{GT}$ is the amount of $CO_2$ produced by $GT$; $D_{GB}$ is the amount of $CO_2$ produced by power $GB$; $\varepsilon_{grid}$ is the $CO_2$ emission coefficient of power grid; $\varepsilon_{GT}$ is the $CO_2$ emission coefficient of $GT$; and $\varepsilon_{GB}$ is the $CO_2$ emission coefficient of $GB$.

*4.2. Optimization Scheduling Constraints*

1.  MMIES Energy Transmission Network Constraints

Based on the mathematical model of the energy transmission network above, the network constraints in the MIES optimization problem are as outlined in Equations (8)–(11).

2.  MMIES Power Balance Constraints

In the MMIES scheduling optimization problem, to make full and reasonable use of the electric energy and natural gas resources of each microgrid while minimizing the total cost as much as possible, the overall power balance between MIESs should be achieved. For each microgrid to carry out their own distributed optimal scheduling, the mutual power among microgrids is introduced, where $P_{i,exc,t}$ represents the mutual power received or sent by microgrid i. Positive values imply received power and negative values imply sent power. After each microgrid is mutualized, the power balance of electric energy will become the following:

$$P_{i,PV,t} + P_{i,WT,t} + P_{i,BSS\_in,t} + P_{i,GT,t} + P_{i,grid,t} - P_{i,BSS\_out,t} - P_{i,EC,t} - P_{i,GSHP\_heat,t} - P_{i,GSHP\_cool,t} - P_{i,PtG,t} - P_{i,sold,t} + P_{i,exc,t} = P_{i,load,t} \tag{31}$$

Among them, the mutual power of each microgrid should also achieve real-time balance:

$$\sum_{i \in N} P_{i,exc,t} = 0 \tag{32}$$

3.  MIES Internal Device Constraints

The types of equipment used in each MIES are roughly similar, and they all need to meet the requirements, respectively. The output of all of the equipment in the energy exchange link and the energy storage link has a certain range, and its power needs to meet the inequality constraints of the upper and lower limits of the equipment output. The formula for specific constraints has been previously described in Equations (1)–(7).

## 5. Distributed Scheduling Based on ADMM

In the scheduling optimization of MMIESs, distributed optimal scheduling architecture is often used to solve the problem of privacy autonomy for each microgrid and complex information interaction among microgrids. The basic architecture for the optimal scheduling established in this paper is shown in Figure 1. Each microgrid belongs to a different decision-making body with autonomy, and optimizes energy resources in the microgrid through the regional control system. The carbon emissions of the entire MMIES are controlled by the overall carbon emissions management system, forming a two-tier scheduling architecture for the economic scheduling of each microgrid at the lower level and for the carbon emissions scheduling of the overall system at the upper level. Through the information exchange between the management systems, the joint cooperative optimization scheduling and control of each microgrid is realized, thus, realizing the distributed cooperative control and joint dispatching of MMIESs with internal autonomy for each subject.

*5.1. ADMM Distributed Optimal Scheduling*

According to the ADMM standard model [25], the constraints between the modular objective function and the subproblems are first determined, and then the optimization

problem is formulated. In actual operation, to ensure independent scheduling and information privacy, each microgrid will not disclose its objective function and equipment operation parameters to other microgrids, so distributed iterative optimization is adopted to achieve the optimal overall economic operation cost. The overall objective expression is as follows:

$$\begin{cases} \min \sum\limits_{i \in N} C_i^{MIES} \\ s.t. \sum\limits_{i \in N} P_{i,exc,t} = 0 \end{cases}, \forall t \in T \tag{33}$$

By adding consistency constraints and using ADMM algorithm decomposition technology, the multi-agent cooperative optimization problem can be decomposed into each microgrid's autonomous scheduling sub-problems. The objective function of a single microgrid can be expressed as follows:

$$\min C_i^{PIES} + \sum\limits_{t \in T} [\lambda_{i,t}(P_{i,exc,t} + \sum\limits_{j \in N, j \neq i} \hat{P}_{j,exc,t}^{CO_2}) + \frac{\rho}{2}(P_{i,exc,t} + \sum\limits_{j \in N, j \neq i} \hat{P}_{j,exc,t}^{CO_2})^2] \tag{34}$$

where $\lambda_{i,t}$ is the Lagrangian multiplier; $\rho > 0$ is the penalty parameter; and $\hat{P}_{j,exc,t}$ is the coordinating variable.

Then, the environmental cost in the carbon emissions management system is optimized, and the overall objective expression is as follows:

$$\begin{cases} \min \sum\limits_{i \in N} C_i^{MIES} + \sum\limits_{i \in N} D_i^{MIES} \\ s.t. \sum\limits_{i \in N} P_{i,exc,t} = 0 \end{cases}, \forall t \in T \tag{35}$$

Its distributed iteration form is as follows:

$$\min \sum\limits_{i \in N} D_i^{PIES} + \sum\limits_{i \in N} \sum\limits_{t \in T} [\lambda_{i,t}(P_{i,exc,t} - \hat{P}_{i,exc,t}) + \frac{\rho}{2}(P_{i,exc,t} - \hat{P}_{i,exc,t})^2] \tag{36}$$

### 5.2. Improved ADMM Distributed Optimal Scheduling Process with Adaptive Step-Size

The MMIES established in this paper contains three microgrids, each of which undergoes an iterative optimization of economic objectives. Before and after the iterative optimization of the three microgrids' economic objectives, the carbon emissions management system optimizes the overall environmental objectives of MMIES, ultimately forming a nested iterative optimization process for the three microgrid economic optimizations in small cycles, and the overall MMIES economic and environmental optimization in large cycles. Compared to single optimization, the iterative interaction of multiple optimizations may affect the computational speed and convergence. Therefore, this paper proposes an adaptive step-size method, which improves the previously used constant penalty parameter $\rho$ to a moderately variable quantity (Figure 3).

The specific solution steps are as follows:

1.  Raw data and equipment parameters, including load data, distributed power output, CHP unit operating parameters, Lagrange multipliers $\lambda_{i,t}$, penalty parameters $\rho$, etc., were inputted.
2.  Each MIES subsystem solved the autonomous optimization problem independently and in parallel for operation optimization. The optimization result $P_{i,exc,t}$ was obtained by solving Equation (34). According to the result, the coordination variable was updated, $\hat{P}_{i,exc,t} = P_{i,exc,t}$ and the obtained coordination variable $\hat{P}_{i,exc,t}$ was used for communication with other MIES and energy nodes.
3.  Considering the influence of carbon emissions generated at the energy node on the overall economy of the region, Equation (36) was solved to obtain the optimization result $P_{i,exc,t}$, and the coordinating variable was updated according to the result $\hat{P}_{i,exc,t}^{CO_2} = P_{i,exc,t}$. The resulting coordination variable $\hat{P}_{i,exc,t}^{CO_2}$ was used to communicate with other MIES.

4.  The coordination variables $\hat{P}_{i,exc,t}^{CO_2}$ are updated according to the optimization results of each MIES. The Lagrange multiplier was then updated with the latest coordination variable:

$$\lambda_{i,t}{}^{[k+1]} = \lambda_{i,t}{}^{[k]} + \rho\left(\hat{P}_{i,exc,t}^{CO_2[k]} - \hat{P}_{i,exc,t}{}^{[k]}\right) \tag{37}$$

5.  The step size was updated:

$$\rho^{[k+1]} = \begin{cases} \dfrac{\rho^{[k]}}{1+\log(\delta_{dual}/\delta_{pri})} & \delta_{dual} < 0.1\delta_{pri} \\ \rho^{[k]}[1+\log(\delta_{pri}/\delta_{dual})] & \delta_{dual} > 10\delta_{pri} \\ \rho^{[k]} & \text{otherwise} \end{cases} \tag{38}$$

6.  If the above was satisfied, then the following was true:

$$\begin{cases} \left\|\hat{P}_{i,exc,t}^{CO_2[k]} - \hat{P}_{i,exc,t}^{[k]}\right\|_2 \leq \delta_{pri} \\ \rho\left\|\hat{P}_{i,exc,t}^{CO_2[k]} - \hat{P}_{i,exc,t}^{CO_2[k-1]}\right\|_2 \leq \delta_{\text{dual}} \end{cases} \tag{39}$$

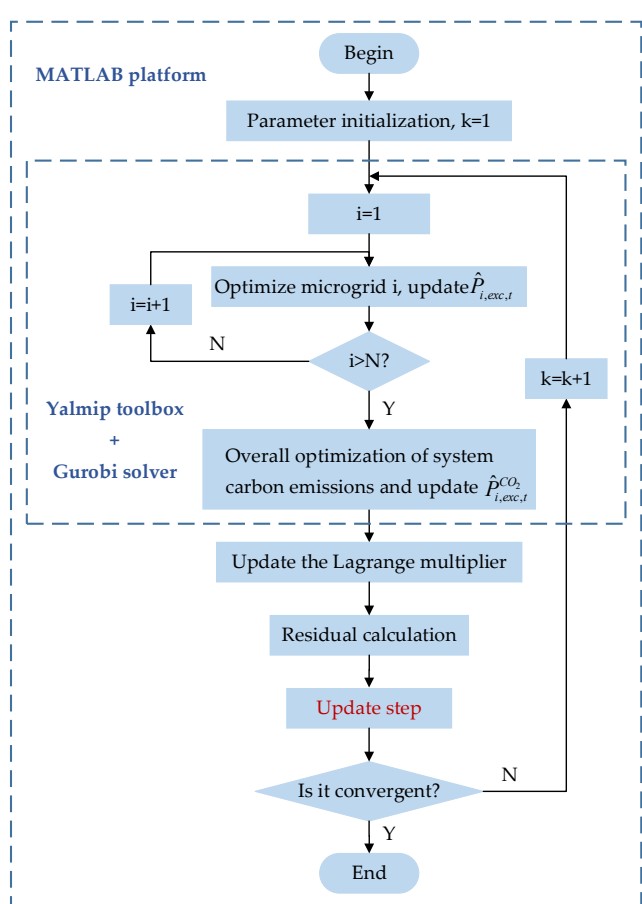

**Figure 3.** Distributed iterative scheduling flowchart.

In this case, the optimal policy scheduling was achieved and $\delta_{pri}$ and $\delta_{dual}$ are the original and dual residuals. Otherwise, let *k = k + 1*, and return to Step 2.

## 6. Example Analysis

The following will be based on the distributed optimal scheduling method based on ADMM proposed in this paper, and the results are compared with the traditional centralized algorithm to verify the calculation accuracy of the distributed ADMM algorithm.

The program calculation is carried out on the MATLAB R2021a platform, using the Yalmip toolbox and Gurobi solver.

### 6.1. Example Basic Data

Based on the operation data and comprehensive energy load data of three MIESs in Zhejiang Province in China, which are the Dongyang residential MIES, Yiwu commercial MIES and Yongkang industrial MIES, this paper constructs an MMIES simulation model, as shown in Figure 1, through the interconnection of energy and gas networks to analyze the operation of the interconnected system. There are slight differences in the capacity of the equipment and the upper limit of the interactive energy of the large grid among the three MIESs. The forecast values of photovoltaic, wind power, and load for each MIES are shown in Figure 4.

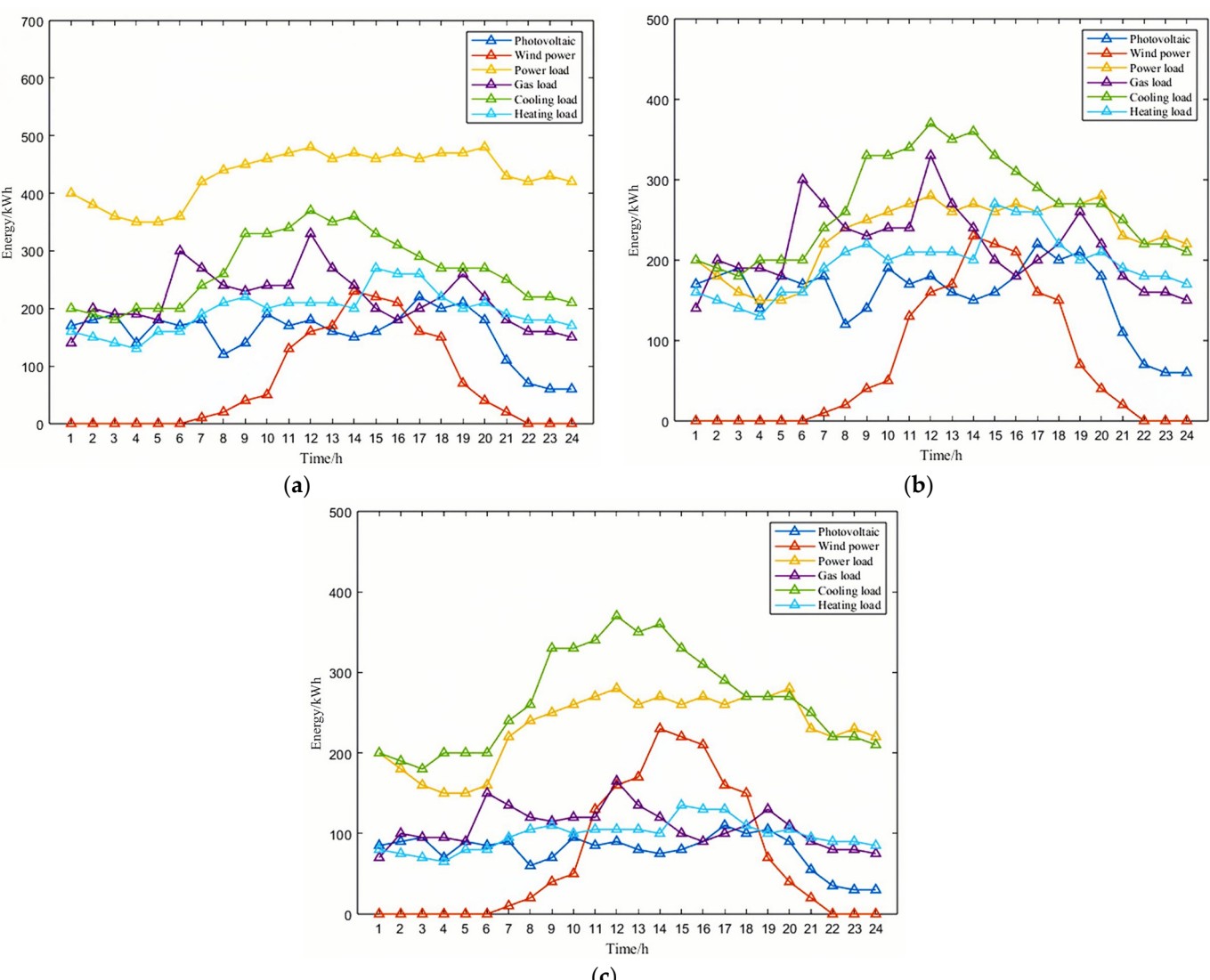

**Figure 4.** Input data of MIESs. (**a**) MIES1; (**b**) MIES2; and (**c**) MIES3.

The equipment parameters of the MIESs are shown in Table 1.

The prices of electricity and natural gas in this region are different at different times of the day, and peak–valley electricity price and peak–valley natural gas price can play a good role in peak cutting and valley filling. The specific prices adopted by the three MIESs are shown in Tables 2 and 3.

In terms of BSS cost calculation, $\alpha$ = 1000 yuan/(kWh), $\beta$ = 3500 yuan/kW, $\gamma$ = 0.1542 yuan/kW, r = 8%, and the BSS life is 10 years.

**Table 1.** Capacity and efficiency of equipment in MIES.

| Equipment | Energy (kWh) and Efficiency of MIES1 | | Energy (kWh) and Efficiency of MIES2 | | Energy (kWh) and Efficiency of MIES3 | |
|---|---|---|---|---|---|---|
| GT | 200 | 0.4 | 100 | 0.4 | 100 | 0.4 |
| GB | 500 | 0.8 | 500 | 0.8 | 500 | 0.8 |
| AC | 100 | 0.7 | 100 | 0.7 | 100 | 0.7 |
| HA | 500 | 0.8 | 500 | 0.8 | 500 | 0.8 |
| EC | 100 | 3 | / | / | / | / |
| GSHP | 100 | 4.5 (cool) 3.5 (heat) | 100 | 4.5 (cool) 3.5 (heat) | 100 | 4.5 (cool) 3.5 (heat) |
| PtG | 100 | 0.55 | 100 | 0.55 | 100 | 0.55 |
| SB | 100 | 0.95 | 200 | 0.95 | 100 | 0.95 |
| Grid | 200 | | 300 | | 500 | |

**Table 2.** Electricity price.

| Time (h) | 1–7 | 8–10 | 11–22 | 23–24 |
|---|---|---|---|---|
| Electricity price (Yuan/kWh) | 0.41 | 0.74 | 1.2 | 0.74 |

**Table 3.** Natural gas price.

| Time (h) | 1–8 | 9–22 | 23 | 24 |
|---|---|---|---|---|
| Natural gas price (Yuan/m$^3$) | 2.63 | 2.05 | 2.63 | 2.05 |

*6.2. Distributed Optimal Scheduling Results of Each MIES*

The operating costs for each MIES are calculated using the distributed algorithm and the centralized algorithm, as shown in Table 4. It shows that for the MMIES established in this paper, the calculation results of the ADMM distributed algorithm and the centralized algorithm are basically consistent, and the error is only 0.0029%. This means that, without sacrificing economic benefits, the decision-making autonomy of MIESs can be realized through the ADMM distributed algorithm, which has almost the same benefit as the centralized scheduling.

**Table 4.** Distributed and centralized optimization results.

| Algorithm | MIES1 | MIES2 | MIES3 | CO$_2$ | Sum |
|---|---|---|---|---|---|
| Centralized (Yuan) | 1647.72 | 1102.38 | 974.32 | 806.05 | 4530.47 |
| Distributed (Yuan) | 1647.70 | 1102.29 | 974.34 | 806.01 | 4530.34 |

The distributed iteration results of each MIES are shown in Figure 5a, where the convergence condition is reached after the 65[th] iteration. The change trend of the results in each microgrid is first rising and then decreasing, and finally tends to the result of centralized optimization. This is because during the initial optimization, the mutual energies $P_{i,exc,t}$ are all close to the direction that is most conducive to the minimum operating cost, and the equilibrium constraint among them $\sum_{i \in N} P_{i,exc,t} = 0$ has not been reflected, so the initial optimization results are all too small. In the later stage, due to the consideration of equilibrium constraints, which are embodied in the objective function in the distributed optimization iteration, there is a temporary increase in the results of each microgrid. In each iteration, the balance between the minimum operating cost of the microgrid and the equilibrium constraint is gradually reached in line with each other. The results converge, the iteration stops, and the target is optimized.

The iteration of the economic cost and carbon emissions-based environmental cost of the entire MMIES are shown in Figure 5b. Through the iterative calculation results, it can be seen that both the optimization results of a single MIES and the overall results of a multi-objective problem converge effectively, indicating the accuracy and reliability of the ADMM in solving this optimization problem.

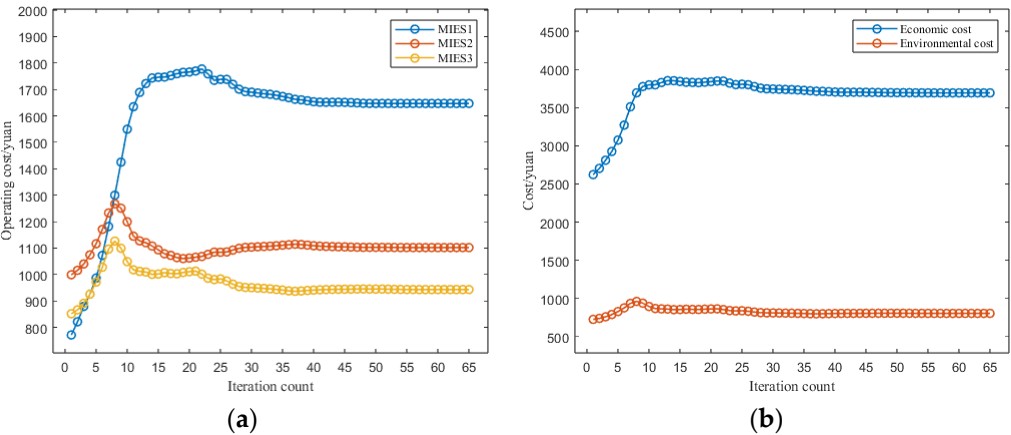

**Figure 5.** Cost iteration results. (**a**) Single MIES cost and (**b**) economic cost and environmental cost.

### 6.3. Energy Interaction between Microgrids

The energy interaction among each microgrid is shown in Figure 6. The mutual energy obtained by the microgrid is positive, and the mutual energy sent out is negative. After iterative convergence, the total mutual energy of each microgrid is basically zero, and the balance is reached. At the same time, due to the constraints of DC power flow and voltage on the lines connecting each microgrid, there are upper and lower limits of mutual energy between microgrids, and very high-power energy interaction does not take place because of cost reduction.

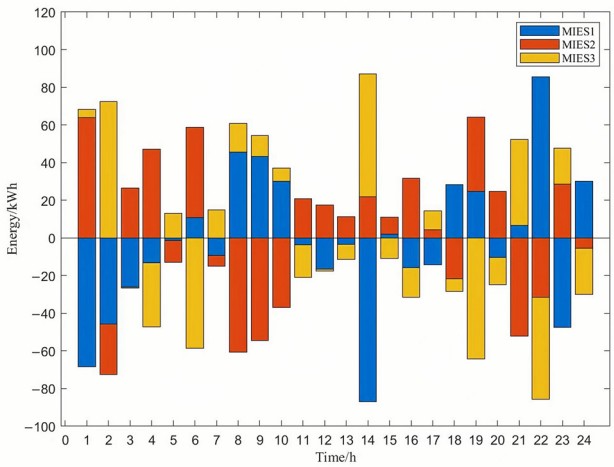

**Figure 6.** Energy interaction between microgrids.

### 6.4. Energy Injection of Each MIES

Figure 7 shows the electric energy injection and gas energy injection of each period, respectively. Due to the transmission energy constraints of the electrical and gas interfaces shown before, the electrical and gas energy are kept within the constrained level, which effectively ensures the safety and stability of the system. The internal energy and gas pipeline constraints in each subregion are similar in the whole system, and the electric energy and gas energy constraints in each subregion also meet their own constraint levels shown in Figure 7. The electricity price is higher at 4:00–8:00, but the user's electricity demand is larger, and the electricity price is lower at 9:00–11:00, 19:00, and 24:00, so the energy obtained by each subregion from the grid is higher during these periods, and most of the other periods are supplied by natural gas to maintain the balance of supply and demand.

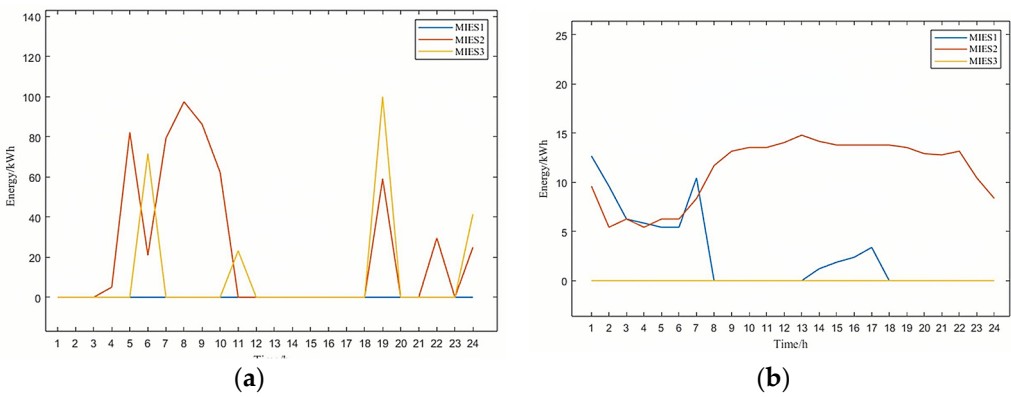

**Figure 7.** Electric energy and gas injection status of MIES. (**a**) Electric energy injection and (**b**) gas injection.

### 6.5. Energy Storage System Operation of Each MIES

Figure 8 reflects the changes in the state of charge of BSSs under different energy storage coordination control strategies. According to different strategies, the final operating costs of microgrids and energy storage construction vary, as shown in the Table 5.

**Figure 8.** Energy storage operation status of MIES: (**a**) without strategy; (**b**) with strategy of two charge and discharge per day; (**c**) with strategy of one charge and discharge per day; and (**d**) with free strategy per day.

**Table 5.** Cost changes with different strategies.

| Energy Storage Coordination Control Strategy | Microgrids Cost (Yuan) | BSS Operation Cost (Yuan) | Total Cost (Yuan) |
|---|---|---|---|
| Without strategy | 4525.47 | 246.34 | 4771.87 |
| One charge and discharge per day | 4561.57 | 181.72 | 4743.29 |
| Two charge and discharge per day | 4526.06 | 219.77 | 4745.83 |
| Free strategy per day | 4530.34 | 210.65 | 4740.99 |

It can be seen that when there is no strategy for charging and discharging, although the operating cost of microgrids is lower, the uncontrolled charging and discharging behavior of BSS leads to large equipment losses, resulting in higher operating costs and overall costs for energy storage construction. Under the strategy of one charge and one discharge or two charges and two discharges of energy storage equipment within a day, the operating cost of microgrids is increased to varying degrees, and the operating cost of energy storage construction is reduced. The free strategy of BSS within each microgrid refers to different devices being able to develop n charging and n discharging strategies according to their own microgrid needs. In this example, the optimization results show that BSS1 executes a one charge one discharge strategy, while BSS2 and BSS3 execute a two charge two discharge strategy, ultimately achieving the lowest total cost of the multi-microgrid system. Compared to being without a strategy, the free strategy per day increased microgrids' cost by 4.87 yuan, an increase of 0.1076%, while BSS operation cost decreased by 35.69 yuan, a decrease of 14.49%, resulting in an overall improvement in the system's economy.

It can be seen in Figure 8 that the charge and discharge of energy storage devices in each MIES is affected by the real-time electricity price and is also related to the renewable energy generation and load demand in the system at the same time. Without the energy storage control, if the wind power and photovoltaic power generation are greater than the load demand from 12:00 to 14:00, the energy storage devices in each MIES start to carry out heavy charging, and the SOC gradually rises from the lowest 0.1 to the highest 0.9. In the following hours, each micro-grid, respectively, carries out small-scale charging and discharging in response to its own load demand to meet the balance of the supply and demand of electric energy in the MIES. Until around 20:00–23:00, the electricity price is at the highest level all day, and the load demand is large, and the energy storage devices begin to discharge on a large scale, reducing the cost of electricity purchase and improving economic benefits.

Under the strategy of two charges and two discharges, the energy storage system can basically complete the initial charging and discharging plan, but the main body of completion may change. For example, when there is no strategy, BSS3 first charges and discharges from 5 to 8 o'clock, and when there is a strategy, BSS2 can replace BSS3 to charge, thereby reducing the overall charging and discharging frequency through cooperation.

Under the strategy of one charge and one discharge, some of the initial charging and discharging plans of the BSS will not be completed, and a small portion of benefits will be sacrificed. However, each BSS only needs to complete one charging and discharging within a day, which can greatly extend the lifespan of the energy storage equipment. Without the energy storage control, such as 13:00–14:00 and 18:00–20:00, there will be some battery charging and some battery discharging phenomena. With the energy storage control some battery charging status is unchanged, while some other batteries charge and discharge on demand, which can not only meet the balance of supply and demand and does not affect the economy of the system, but also avoids unnecessary battery charging and discharging phenomena. Reducing the frequency of charge and discharge is beneficial to the operation of the system.

Based on the differences in optimization results achieved by these different strategies, each microgrid entity can choose different strategies according to their own needs, making the overall optimization of microgrid clusters more flexible.

### 6.6. Effect of Carbon Emissions Weight on Overall Optimal Scheduling of MMIES

In order to further explore the relationship between the two objective functions and explain the impact of the penalty factor $\lambda_D$ of the $CO_2$ emissions on the system low-carbon economy, $\lambda_D$ was set as 0, 0.5, 1, 1.5, 2, 2.5, 3, 3.5, and 4 times, respectively, and the changing trends of operating cost and carbon emissions environmental cost under different weights are obtained in Figure 9. It can be clearly seen that with the increase of the penalty on $CO_2$ emissions in the upper coordination system, $CO_2$ emissions gradually decreased and the environmental cost of carbon emissions gradually decreased, while the operating cost increased. After 2.5 times the weight, the trend slowly decreased until it no longer changed. This is because the reduction of $CO_2$ emissions was achieved by changing the ratio of electricity and gas energy supply, but under the constraints of various energy supplies in the system, the ratio of the two had reached the limit, so it could not be further changed to meet the load demand of the system. For different systems, the penalty factor can be reasonably selected according to local carbon emissions policies and the needs of decision makers to achieve different overall scheduling optimization effects.

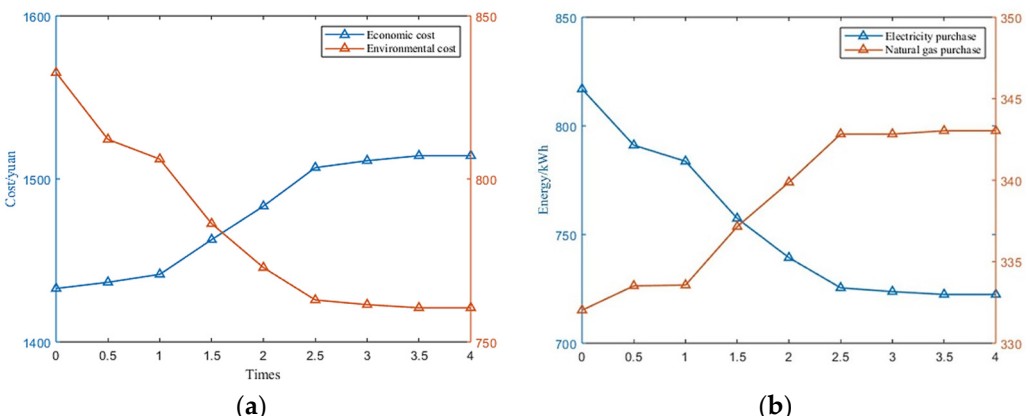

**Figure 9.** Cost, electricity, and natural gas purchase changes under different penalty factors. (**a**) Cost changes. (**b**) Electricity and natural gas purchase changes.

As can be seen from Figure 9b, with the increase in the carbon emissions penalty factor $\lambda_D$, the purchase of electricity gradually decreased, while the purchase of gas gradually increased. This is because the carbon emissions caused by grid energy generation are more damaging to the environment than gas turbines and gas boilers in MIES, so for different needs of the system's carbon emissions regulation, the system can choose its own equipment capacity.

Through these comparisons, it can be seen that adding a carbon emission system to MMIES has a significant impact on both cost and the proportion of electricity and natural gas purchases, which also changes the degree of the environmental impact of the system. The carbon emissions system can effectively monitor the carbon emissions of the system, make changes in punishment strategies based on the environment, and greatly improve the environmental friendliness without excessively conceding benefits, which contributes to sustainable development issues.

### 6.7. Convergence Analysis Based on Adaptive Step-Size

To solve the problem of increasing computational difficulty, this paper improved the model based on single-objective distributed iterative calculation and proposed a distributed iterative optimization method based on the adaptive step-size method. Based on an unchanged original step size, the step size would be trimmed to a certain extent according to the calculation situation to improve the computational efficiency and reduce the number of iterations as much as possible. Figure 10 shows the convergence properties of distributed optimization with and without the adaptive step-size method. Table 6 provides the specific convergence time and convergence frequency.

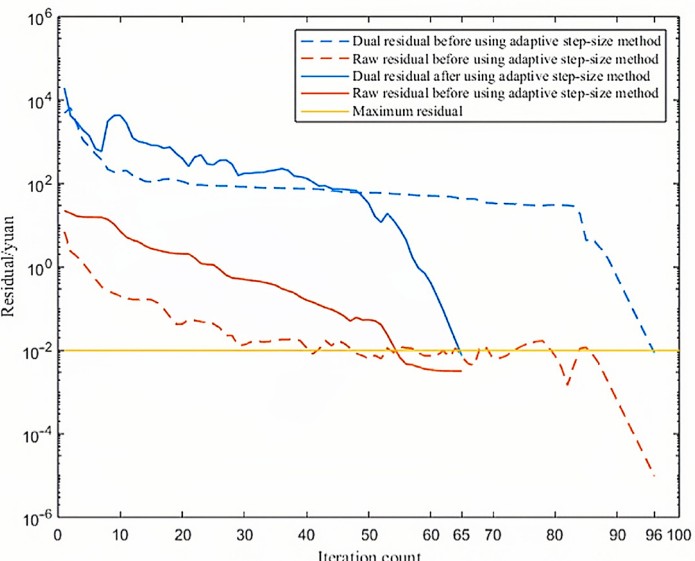

**Figure 10.** Iterative residual variation.

**Table 6.** Iterative residual variation.

| Step | Constant Step | Adaptive Step-Size |
|---|---|---|
| Calculate the number of iterations (time) | 96 | 65 |
| Calculation time (s) | 1316 | 1181 |

Dashed lines show the iterative convergence of distributed computing for each MIES without using the adaptive step-size method. Here, the maximum residual limit is taken to be $10^{-2}$. The original residual decreases faster, and at the 40th calculation, the convergence condition lingers at the edge of the convergence condition, while the dual residual decreases more slowly. Finally, at the end of the 96th calculation, the dual residual of the original residual meets the convergence condition at the same time, and the final optimization result is obtained after the calculation is completed.

Solid lines show the iterative convergence of distributed computing for each MIES using the adaptive step-size method. Compared to iterating at the rated step-size, the descent speed of the original residual has been significantly improved, and the overall convergence condition has been reached after the 65th iteration, reducing the number of iterations by 32.3%, and the calculation time reduced by 10.3%. The improved adaptive step-size distributed iteration method in this paper can effectively reduce the number of iterations and improve computational efficiency.

## 7. Conclusions

In this paper, several methods of distributed iterative optimization for MMIESs are analyzed, and the ADMM method based on adaptive step-size was selected for computational efficiency. In this paper, the basic framework of MMIESs containing cold, heat, electricity, and gas energy was established, and a coordinated control strategy limiting the frequency of the charge and discharge of energy storage devices was proposed. Then a multi-objective optimization model based on operating cost and environmental cost for carbon emissions was established. Considering that different microgrids may be managed by different operators and the convergence speed of multi-objective optimization iteration, an adaptive step-size distributed iterative optimization method was used based on ADMM, which can effectively reduce the cost and protect the privacy of each microgrid. Finally, a system composed of three microgrids was provided as an example for simulation analysis, and the following conclusions can be drawn from the analysis:

For the MMIES established in this paper, the simulation results of the ADMM distributed algorithm and the centralized algorithm are basically consistent, and the error

is only 0.0029%, which means that the decision-making autonomy of the MIES can be realized through the ADMM distributed algorithm without sacrificing economic benefits, and the benefits of the distributed scheduling are almost the same as those of the centralized scheduling.

In the multi-objective optimal scheduling results, the interactive energy and energy injection between each microgrid meet the constraints, and the energy storage is also utilized under the constraints of the control strategy, which can not only cut the peak and fill the valley, with BSS operating cost decreasing by 14.49%, but also does not cause too much damage to the energy storage devices, ensuring a stable, economic, low-carbon, and environmentally friendly operation of the system.

Considering the multi-objective optimization of the environmental cost of carbon emissions, with an increase in the penalty factor, carbon emissions can be effectively reduced, but operating costs will also increase, which can provide a reference for decision makers.

The improved adaptive step-size distributed iterative optimization method will trim the step-size according to the calculation situation based on the original step-size, so as to improve the calculation efficiency and reduce the number of iterations as much as possible, reducing the number of iterations by 32.3%, and reducing the calculation time by 10.3%.

In follow-up research, MMIESs should also take into account the volatility of distributed renewable energy generation and the uncertainty of various regional loads, and take such factors into consideration in the model and method for optimization so as to improve the optimal scheduling problem. The current MIESs are connected to the same voltage node, and in the future, addressing the power flow problem for cases in which MIESs are connected to different voltage nodes would be an interesting research direction.

**Author Contributions:** Conceptualization, L.S., Z.C., Y.L., and F.W.; methodology, L.S., Z.C., and Y.L.; software, Z.C.; formal analysis, L.S. and Z.C.; Investigation: K.L. and D.Y.; data curation: L.S. and Z.C.; writing—original draft preparation, L.S., Z.C., Y.L., and F.W.; writing—review and editing, L.S., Z.C., Y.L., F.W., K.L., and D.Y.; supervision, F.W. All authors have read and agreed to the published version of the manuscript.

**Funding:** This research was funded by the State Key Laboratory of Smart Grid Protection and Operation Control of China and the National Natural Science Foundation of China, grant number 52061635102; the Natural Science Foundation of China, grant number 52107088; and the Natural Science Foundation of Jiangsu Province, grant number BK20210365.

**Institutional Review Board Statement:** Not applicable.

**Informed Consent Statement:** Not applicable.

**Data Availability Statement:** We kindly request that any inquiries regarding the data used in this study be directed to the corresponding author. The data sets used in this study are available upon request.

**Conflicts of Interest:** Author Dongmei Yang was employed by the company Nari Group Corporation. The remaining authors declare that the research was conducted in the absence of any commercial or financial relationships that could be construed as a potential conflict of interest.

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
