# Peer review of "Distributed Optimization of Multi-Microgrid Integrated Energy System with Coordinated Control of Energy Storage and Carbon Emissions"

_sustainability, doi:10.3390/su16083225_

Round 1

Reviewer 1 Report

Comments and Suggestions for Authors

The topic concerning the multi-microgrid integrated energy system is interesting. In order to improve the quality of this paper, some suggestions are given  as follow.

1. In this paper, there are so many equations given directly, whether some references are needed to give the reason of why?

2. In Section 5, the scheduling depends on the optimal objectives. How to define the optimization function depends different factors. But as the above question, this paper gives the optimization functions directly and has no reference.

3. The analysis of the simulation results in Section 6 is a bit simple, just giving the data table and few words. Whether the simulation analysis can be more specific and detailed?

4. In conclusion, what's the further work? It is a interesting suggestion.

Comments on the Quality of English Language

The quality of this paper is well.

Reviewer 2 Report

Comments and Suggestions for Authors

Dear authors,

The manuscript “Distributed optimization of multi-microgrid integrated energy system with coordinated control of energy storage and carbon emission” and number: sustainability-2929776 is suitable for publication in your journal after taking into account the following notes:

Keywords are inappropriate and contain repetition.

Figure 4, and Figure 10 are not clear.

Line 139: I think the term “Heat Absorption (HA)” is not used in scientific sources.

Lines 144 and 147 use the word power: the correct word is energy.

Figure 4, Figure 6, Figure 7, and Figure 9.b. Please use energy, not power, and also in the explanation within the text.

Table 1 Please use energy, not capacity.

Table 2: Energy price per kWh and not power per kW.

Please add some results and indicators for the case study to the conclusion.

Please add a table of abbreviations or a minimum definition of the abbreviation in the text when used for the first time; for example, battery storage system (BSS) 168, 71

Best wishes,

Comments on the Quality of English Language

 Minor editing of English language required

 For example, in the title, it is preferable to use emissions  instead of emission

Reviewer 3 Report

Comments and Suggestions for Authors

The paper is good for publication. It has a well-defined structure, plots and results. More validation is needed about YALMIP and Gurobi implementation to improve the accuracy of the results. The conclusion is written in a compact form. Avoid the numbering. Also, you must write the paper according to the template of the journal. Leave space for example line 446. Connect the flowchart with YALMIP and Gurobi optimizers. Characterize the optimization problem, how it is solved and how the plots are derived. Also, mention the solution algorithm. Also, explain the convergence of the algorithm to the solution. Why a conclusion is not required. (line 540). Declare the robustness of your method.

Comments on the Quality of English Language

The paper needs editing

Round 2

Reviewer 3 Report

Comments and Suggestions for Authors

This reviewer read the response letter and the revised manuscript. No other issues had been detected. No further concerns.